# IMPA versus Cloud Analysis and IDA: Different Methods to Evaluate Structural Seismic Fragility

**Carlotta Pia Contiguglia** [1,*], **Angelo Pelle** [1], **Bruno Briseghella** [2] **and Camillo Nuti** [1]

1   Department of Architecture, Roma Tre University, 00153 Rome, Italy; angelo.pelle@uniroma3.it (A.P.); camillo.nuti@uniroma3.it (C.N.)
2   College of Civil Engineering, Fuzhou University, Fuzhou 350108, China; bruno@fzu.edu.cn
*   Correspondence: carlottapiacontiguglia@uniroma3.it

**Abstract:** Well-known methods for seismic performance assessment, such as incremental dynamic analysis (IDA), multi-stripes analysis (MSA) and the cloud method, involve nonlinear response time-history analyses to characterize the relationship between the chosen damage measure versus intensity measure. Over the past two decades, many authors have proposed simplified procedures or nonlinear static approaches to develop fragility. In these procedures, the capacity of the system is evaluated by nonlinear static procedures (i.e., the capacity spectrum method (CSM), the N2 method, modal pushover analysis (MPA)) and the demand is derived by response spectra. In addition to the familiar ones, incremental modal pushover analysis (IMPA) is a novel nonlinear static procedure proposed in recent years, and it is used in this research to present an *IM*-based fragility estimation. The accuracy and effectiveness of different methods to assess vulnerability are investigated by comparing fragility curves derived by MPA-based cloud analysis, IMPA and cloud analysis against IDA. The comparison gives valuable insights on the influence of scaling on different sets of records; however, a more extended validation is needed to confirm the obtained results and draw more general conclusions. Results arise from two relatively small bins of record motions differing by ranges of Joyner-Boore distance and scattered in a range of magnitude are presented.

**Keywords:** IDA; the cloud method; IMPA; MPA; nonlinear static analysis; nonlinear dynamic analysis; fragility curve

## 1. Introduction

Performance-based earthquake engineering (PBEE) procedures allow the prediction and evaluation of the probabilistic seismic performance of bridges and buildings in terms of system-level decision variables, such as loss of use, repair cost and casualties. In the United States, the first generation of PBEE assessment and design procedures for buildings (SEAOC Vision 2000, FEMA 273, ATC-40 [1–4]) took significant steps toward achieving performance-based earthquake engineering. Since then, the Pacific Earthquake Engineering Research Center (PEER) has been working on developing a more robust methodology that involves four stages: hazard analysis, structural analysis, damage analysis, and loss analysis [5]. In the third stage, damage analysis, fragility functions describe the conditional probability of component, element or system to be damaged for a given intensity measure. The first attempt to determine fragility curves can be dated back to 1975, when the Seismic Design Decision Analysis (SDDA) procedure was proposed in the US [6]. Further developments [7,8] were initially applied in the field of the nuclear industry to define a probabilistic relationship between an intensity measure representing seismic input and a damage measure representing the seismic failure of a component of a nuclear power plant. From then on, several methods to estimate fragility (expert-based, experimental, analytical, hybrid, empirical) have been developed by researchers worldwide, relying on different assumptions and restrictions to overcome prevalent intrinsic uncertainties. However, due

to the extremely high subjectiveness, lack of data and other drawbacks typical of expert-based, empirical and experimental methods, the common practice has aroused its interest in analytical and hybrid methods during the last two decades.

Among analytical approaches to derivate fragility curves (probabilistic seismic demand model, elastic spectral analysis, nonlinear static analysis, linear or nonlinear time history analysis [9–17]), incremental dynamic analysis (IDA) is a parametric analysis method developed in 1998 and deeply discussed in 2002 [18,19]. IDA became a worldwide method used by engineers and researchers, and it is still widespread. However, the introduction of uncertainties due to an excessively coarse description of seismic input with varying intensity is inevitable. Eventually, many authors have pointed out that a simple amplitude scaling of ground motion records is one of the main shortcomings in IDA, together with its high computational demand [20–23]. In IDA, a certain number of inputs amplitude scaled to define $IM = im$, then NL-THA is performed and $DCR_{LS}$ so determined are used to define the distribution of $DCR_{LS} | IM = im$. This process is repeated by varying the scale factor to define the seismic response in a whole range of seismic intensities. According to previous research [24–26], the median response of a structure subjected to scaled ground motion records is comparable to that of a structure subjected to unscaled earthquake ground motion records. However, because a single $IM$ is a highly simplified description of ground-motion severity, the value of $DCR_{LS}$ from different ground-motion with $IM = im$ may be different, and thus the relationship $DCR_{LS} \sim f(IM)$ is probabilistic. A common limitation in current databases is the lack of strong ground motion records covering high-intensity intervals at specific periods of the structure [27,28]. Thus, an excessive scaling to fit within high-intensity intervals may occur, biasing the structural response [29], generating a false correlation between $IM$s and EDPs, and increasing uncertainties in the structural response.

In contrast to IDA and MSA (multiple-stripe analysis), the cloud method [26,30,31] involves nonlinear analysis of the structure subjected to a sample of different distance/intensity combined unscaled as-recorded ground motion, which may reduce the number of analyses, uncertainties in seismic input with intensity, computational effort in defining a seismic fragility curve [32], and is based on a regression in the logarithmic space of structural response versus seismic intensity.

To respond to the need for simplified, faster and/or approximate methods, from the late 1990′s, many studies have been published regarding the use of pushover analysis procedures to assess seismic vulnerability, mostly on bridges. In this case, the capacity of the system is evaluated by using nonlinear static procedures (NSPs) (i.e., the capacity spectrum method [33–36], the N2 method [37–39], and modal pushover analysis [20]), while the demand is estimated by response spectra. To assess the reliability of these analytical procedures, different authors have compared developed fragility curves to those obtained by nonlinear time history analysis [34,36,38].

Among NSPs, a novel procedure called incremental modal pushover analysis (IMPA) has been proposed in recent years by Bergami and his co-workers [40]. IMPA requires the execution of modal pushover analysis (MPA) and the evaluation of structural performance within a range of different seismic intensity levels to develop a multimodal capacity curve in terms of base shear versus top displacement. This approach is suitable for performing a displacement-based design procedure and structural analysis of existing structures, yet authors have not suggested the analytical estimation of fragility.

This paper aims to evaluate the reliability of structural fragility derived by the methods mentioned, advancing an $IM$-based derivation of structural fragility, strikingly similar to IDA, based on IMPA. It is known that IDA has a small sensitivity to record-to-record variability compared to other methodologies. Nevertheless, the results indicate that, amidst its slightly higher sensitivity, IMPA has the advantage of requiring considerably smaller computational effort to perform the structural analysis.

The authors argue that scaling response spectra at a higher range of intensities might introduce less uncertainties than a simple amplitude scaling of ground motions. Further

steps of this research will address how the uncertainties in the seismic input affect the reliability of IMPA versus IDA seismic fragility for strong ground motions.

In the following paragraph, analyses are carried out on a real RC frame belonging to a school building located in Norcia (Italy). Each nonlinear dynamic and static procedure is briefly presented, including a step-by-step computational procedure of IMPA. Finally, the reliability of pushover-based estimation of seismic vulnerability is assessed by comparing these curves to those obtained by IDA.

## 2. Methodology

### 2.1. Choice of Engineering Demand Parameter and Intensity Measure

The first-mode spectral acceleration $S_a(T_1, \xi = 5\%)$ is commonly used as an intensity measure (*IM*) parameter [20,21]. Shome et al. [24] stated that the nonlinear response of an MDOF structure dominated by the first mode of vibration depends on the "intensity" of the records at the first period of vibration, while magnitude and distance play a minor role in it. For these reasons, the 5% damped spectral acceleration at the structure's first-mode period $S_a(T_1, \xi = 5\%)$ or simply $S_a$ is adopted as the *IM* in this work, since the structure selected as the case study is dominated by the first mode of vibration (structure's first-mode period of vibration $T_1 = 0.62$ s and the modal mass participation at first-mode is 82%, see Section 3.1 for more details).

In the literature, various engineering demand parameters (EDPs) have been proposed [24], somehow representative of the structure's local or global damaged state. In this study, the critical demand to the capacity ratio for the desired limit state (*LS*), denoted as $DCR_{LS}$ [23,41], is assumed to be the EDP. It represents the demand-to-capacity ratio which brings the system closer to the onset of limit state (herein, the life-safety limit state). The weakest-link formulation is adopted to evaluate the $DCR_{LS}$ (Equation (1)), which means that if the demand-to-capacity ratio $D_{jl}/C_{jl}$ is equal to or higher than unity in just one element, then the structure attains the expected limit state for the *l*th mechanism.

$$DCR_{LS} = \max_l^{N_{mech}} \max_j^{N_e} \left( \frac{D_{jl}}{C_{jl}(LS)} \right) \tag{1}$$

where $N_{mech}$ and $N_e$ are the numbers of the considered potential mechanism of failure and the number of the elements taking part in the *l*th mechanism, respectively. $D_{jl}$ and $C_{jl}(LS)$ are the demand and the limit state capacity, respectively, evaluated for the *j*th element of the *l*th mechanism.

In particular, in this work, only a ductile failure mechanism in columns and beams is considered as a potential failure mechanism ($N_{mech} = 1$). Therefore, in this deformation-based critical $DCR_{LS}$, the demand $D$ is expressed in terms of maximum chord rotation in the *j*th component. Instead, the capacity $C$ in terms of chord rotation is evaluated according to guidelines reported in the Commentary [42] of NTC 2018. Namely, the chord rotation for life-safety limit state is defined as $\frac{3}{4}$ of that corresponding to near-collapse limit state $\theta_u$, evaluated according to Equation C8.7.2.5 of the Commentary (Equation (2)).

$$\theta_u = \left( \theta_y + (\phi_u - \phi_y) L_{pl} \left( 1 - \frac{0.5 L_{pl}}{L_v} \right) \right) \tag{2}$$

where $\theta_y$ is the yield chord rotation, $\phi_u$ and $\phi_y$ are the ultimate and yield curvature, respectively, $L_{pl}$ is the plastic hinge length, and $L_v$ is the shear length.

### 2.2. Record Selection

The PEER Next Generation Attenuation (NGA)—West 2 Project [43] has been used to define the main database of 210 ground motions. In addition, two ground motions representing the 2016 Norcia earthquake, with epicentral distances of 4.6 km and 26.9 km,

respectively, have been extracted from the Italian Accelerometric Archive [44] and included in the aforementioned database.

A set of 36 as-recorded ground motions listed in Table 1 has been defined with an average shear wave velocity to a depth of 30 m ($V_{s,30}$) falling between 213 m/s and 724 m/s, therefore corresponding to the types of mass B′ or C′ (according to Eurocode 8 [45]) which are mixed into the set. This latter comprehends highly scattered values of magnitude $M_w$, ranging from 5 to 7.5, and Joyner-Boore distance included between 0 km and 50 km. The set includes about 50% of near-fault (19 records), ranging from an epicentral distance of 0–10 km, and 50% of far-field records (17 records), or records with an epicentral distance greater than 10 km. The selection exhibits a prevalence of three fault mechanisms: normal, reverse, and strike-slip. Since the frame model used in this study is 2D (see Section 3), two orthogonal directions of the same seismic event are avoided. Record selection also comprehends a wide range of *IM* and distributed values of *DCR$_{LS}$*, with at least one-third of the values greater than 1 [23]. The original set of 36 records was split into two subsets depending on the epicentral distance as shown in Figures 1 and 2, and these were studied separately to investigate the different effects of near-fault versus far-field ground motions. It is well known that the proximity to the fault renders the same ground motions (NF) different from ordinary (FF) ground motions [46]. The near-fault records selection avoids including pulse-like ground motions. Impulsive signals have been identified using the open-source algorithm proposed by Shahi and Baker [47,48]. This can identify pulses at arbitrary orientations using continuous wavelet transforms of two horizontal orthogonal components of a ground motion to identify the orientation that may contain a pulse.

**Table 1.** Details of the two subsets of ground motion data deepened for the study based on the NGA—West 2 database.

| File ID | Earthquake Name | RSN | Year | Mech. | $M_w$ | $R_{jb}$ [km] | $V_{s,30}$ [m/s] | DS-595 [s] | DS-575 [s] |
|---|---|---|---|---|---|---|---|---|---|
| 1 | "Oroville-01" | 106 | 1975 | Normal | 5.89 | 7.79 | 680.37 | 3.4 | 1.5 |
| 2 | "Oroville-03" | 114 | 1975 | Normal | 4.7 | 7.35 | 418.97 | 4.4 | 1.3 |
| 3 | "Santa Barbara" | 136 | 1978 | Reverse Oblique | 5.92 | 0 | 514.99 | 7.5 | 4.3 |
| 4 | "Tabas_Iran" | 139 | 1978 | Reverse | 7.35 | 0 | 471.53 | 11.3 | 6.7 |
| 5 | "Helena_Montana-01" | 1 | 1935 | Strike-slip | 6 | 2.07 | 593.35 | 2.5 | 1.2 |
| 6 | "Dursunbey_Turkey" | 144 | 1979 | Normal | 5.34 | 5.57 | 585.04 | 2.5 | 1.4 |
| 7 | "Coyote Lake" | 145 | 1979 | Strike-slip | 5.74 | 5.3 | 561.43 | 8.5 | 2.7 |
| 8 | "Norcia_Italy" | 156 | 1979 | Normal | 5.9 | 1.41 | 585.04 | 5.7 | 2.7 |
| 9 | "Livermore-02" | 222 | 1980 | Strike-slip | 5.42 | 7.94 | 550.88 | 4.5 | 1.1 |
| 10 | "Anza (Horse Canyon)-01" | 226 | 1980 | Strike-slip | 5.19 | 5.85 | 617.78 | 2.4 | 1.1 |
| 11 | "Mammoth Lakes-06" | 249 | 1980 | Strike-slip | 5.94 | 6.45 | 373.18 | 5.1 | 2.5 |
| 12 | "Izmir_Turkey" | 134 | 1977 | Normal | 5.3 | 0.74 | 535.24 | 1.6 | 0.3 |
| 13 | "Mammoth Lakes-07" | 253 | 1980 | Strike-slip | 4.73 | 3.86 | 377.41 | 10.2 | 3.1 |
| 14 | Imperial Valley-02 | 6 | 1940 | Strike-Slip | 6.95 | 6.09 | 213.44 | 24.2 | 17.7 |
| 15 | Chalfant Valley-04 | 563 | 1986 | Strike-Slip | 5.44 | 8.88 | 316.19 | 17.1 | 7.7 |
| 16 | Kalamata, Greece-01 | 564 | 1986 | Normal | 6.2 | 6.45 | 382.21 | 6.1 | 1.9 |
| 17 | Kalamata, Greece-02 | 565 | 1986 | Normal | 5.4 | 4 | 382.21 | 4.2 | 1 |
| 18 | Loma Prieta | 752 | 1989 | Reverse Oblique | 6.93 | 8.65 | 288.62 | 13.2 | 5.6 |
| 19 | Central Italy | n.a. | 2016 | Normal | 6.5 | 4.6 | 498 | n.a. | n.a. |
| 20 | "Kern County" | 15 | 1952 | Reverse | 7.36 | 38.42 | 385.43 | 30.3 | 10.7 |
| 21 | "Lytle Creek" | 49 | 1970 | Reverse Oblique | 5.33 | 42.14 | 667.13 | 5.1 | 2.9 |
| 22 | "Santa Barbara" | 135 | 1978 | Reverse Oblique | 5.92 | 23.75 | 465.51 | 7 | 3.4 |
| 23 | "San Fernando" | 81 | 1971 | Reverse | 6.61 | 35.54 | 529.09 | 13.7 | 7.1 |
| 24 | "Northern Calif-07" | 101 | 1975 | Strike-slip | 5.2 | 28.73 | 567.78 | 5.7 | 4.3 |
| 25 | "Oroville-02" | 108 | 1975 | Normal | 4.79 | 12.07 | 377.25 | 7.1 | 3.3 |
| 26 | "Friuli_ Italy-01" | 125 | 1976 | Reverse | 6.5 | 14.97 | 505.23 | 4.9 | 2.5 |
| 27 | "Coyote Lake" | 152 | 1979 | Strike-slip | 5.74 | 20.44 | 362.98 | 8.2 | 3.9 |
| 28 | "Norcia_ Italy" | 157 | 1979 | Normal | 5.9 | 13.21 | 535.24 | 10.5 | 5.9 |
| 29 | "Anza (Horse Canyon)-01" | 225 | 1980 | Strike-slip | 5.19 | 12.24 | 724.89 | 2.1 | 0.7 |
| 30 | "Victoria_ Mexico" | 265 | 1980 | Strike-slip | 6.33 | 13.8 | 471.53 | 8.2 | 4.4 |
| 31 | "Mammoth Lakes-04" | 241 | 1980 | Strike-slip | 5.7 | 12.75 | 537.16 | 11.5 | 3.4 |
| 32 | "Mammoth Lakes-09" | 274 | 1980 | Strike-slip | 4.85 | 10.96 | 377.41 | 16 | 7.7 |
| 33 | "Almiros_ Greece" | 279 | 1980 | Normal | 5.2 | 13.25 | 412.68 | 10 | 4.6 |
| 34 | "Coalinga-02" | 370 | 1983 | Reverse | 5.09 | 24.23 | 467.03 | 13.7 | 8.6 |
| 35 | "Borah Peak_ ID-02" | 442 | 1983 | Normal | 5.1 | 16.31 | 468.44 | 5 | 2.3 |
| 36 | Central Italy | n.a. | 2016 | Normal | 6.5 | 26.9 | n.a. | n.a. | n.a. |

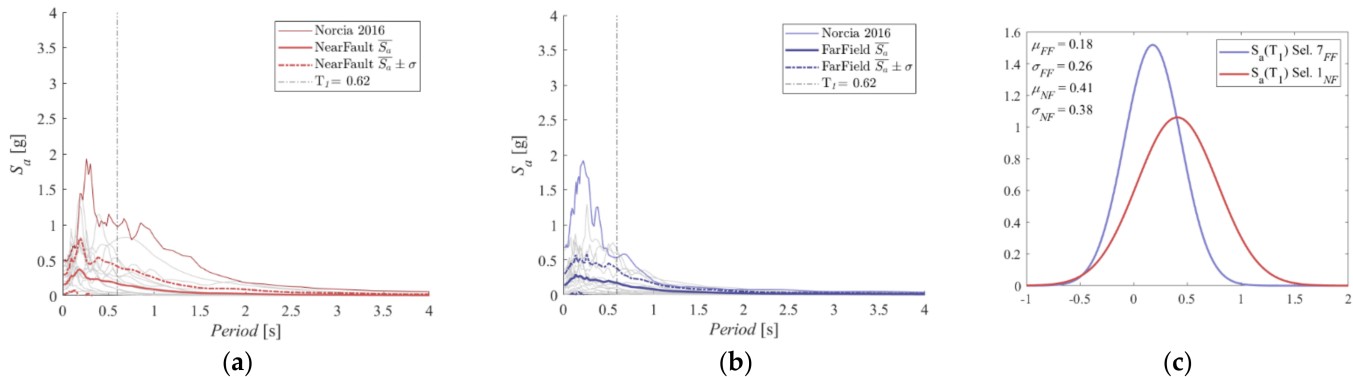

**Figure 1.** Elastic response spectra of the (**a**) near-fault and (**b**) far-field subset of records. The figures highlight the two records of the 2016 Norcia Earthquake, $\overline{S_a}$ is the average response spectra from the two set, and $\overline{S_a} \pm \sigma$ is the range of variance according to standard deviation. (**c**) Normal distribution of $S_a$ for $T = T_1$.

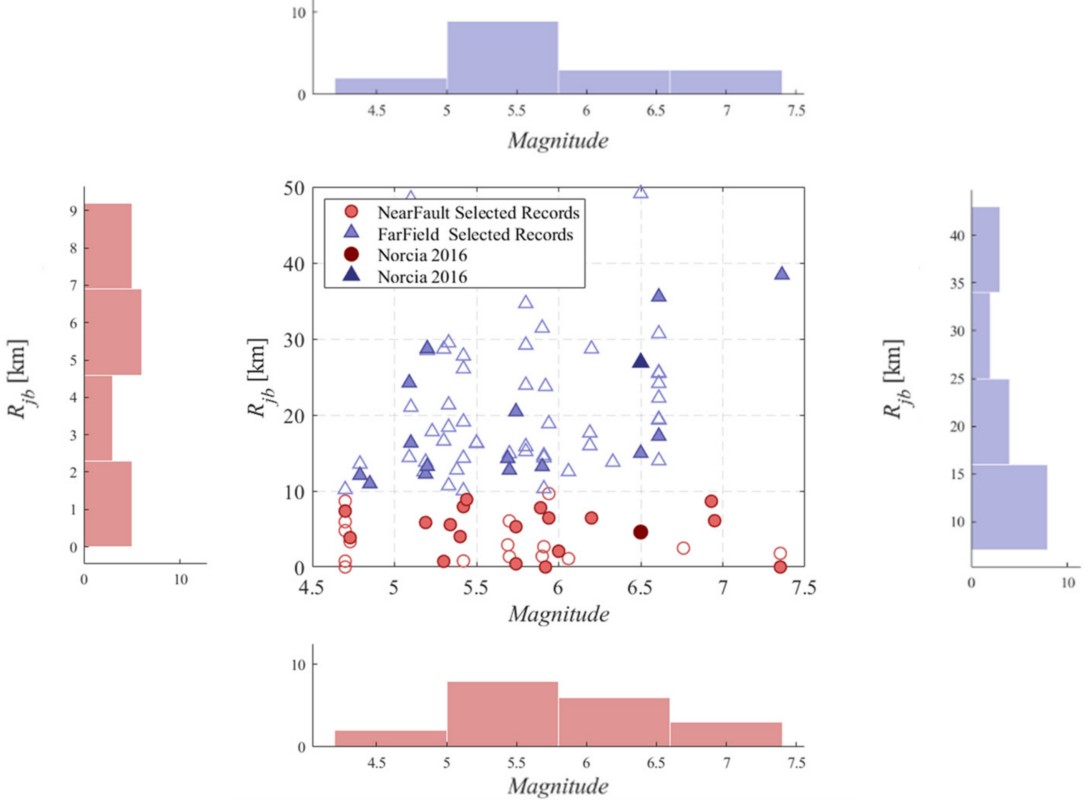

**Figure 2.** $M_w$ magnitude–$R_{jb}$ distance scatter diagrams of the two subsets, Sel. 1 $_{NF}$ and Sel. 7 $_{FF}$.

### 2.3. Performed Nonlinear Analysis

#### 2.3.1. Cloud-Based Analysis

The cloud-based Analysis (CA) is particularly suitable to assess structural fragility both for the simplicity of its formulation and for the low required computational effort. Conversely, it is extremely sensitive to the record selections and based on a few simplifying assumptions, such as fixed standard error of the regression [23,31,32,39,41,49].

CA adopts a linear regression model in the logarithm scale to fit the pairs of demand to capacity ratio ($DCR_{LS}$) and *IM*, where $DCR_{LS}$ are calculated through nonlinear analysis. The regression-based probability model describes the $DCR_{LS}$ for a given *IM* level and can be evaluated by Equations (3) and (4):

$$\mathrm{E}[\ln DCR_{LS}|IM] = \ln \eta_{DCR_{LS}|IM} = \ln a + b \ln IM \tag{3}$$

$$\sigma_{\ln \beta_{DCR_{LS}|IM}} \cong \beta_{DCR_{LS}|IM} = \sqrt{\sum_{i=1}^{N} \left( \ln DCR_{LS,i} - \ln \eta_{DCR_{LS}|IM_i} \right)^2 / (N-2)} \tag{4}$$

where $\mathrm{E}[\ln DCR_{LS} | IM]$ is the expected value for the natural logarithm of $DCR_{LS}$ given *IM*, and $\eta_{DCRLS|IM}$ and $\sigma_{\ln DCRLS|IM}$ are the median and logarithmic standard deviation for $DCR_{LS}$ given *IM*, respectively. The constants ln*a* and *b* are the linear least square regression coefficients. Finally, the structural fragility obtained based on the CA is (Equation (5)):

$$\mathrm{P}(DCR_{LS} > 1|IM) = \mathrm{P}(\ln DCR_{LS} > 0|IM) = \Phi\left( \frac{\ln \eta_{DCR_{LS}|IM}}{\beta_{DCR_{LS}|IM}} \right) \tag{5}$$

where *Φ* is the standard Gaussian cumulative distribution function.

In this work, two different methodologies are adopted to find the relationships of *IM* versus $DCR_{LS}$ for the structure under investigation. Namely, in one case, time-history analyses are employed to evaluate the demand $D_{jl}$ (demand of the *j*th element of *l*th mechanism) at each time step. Within this paper, this approach is named dynamic cloud analysis, or shortly D-CA.

In the other case, the demand $D_{jl}$ is computed by adopting the modal pushover analysis (MPA). This approach is referred to as MPA-CA. The modal pushover analysis [50,51] is a nonlinear static procedure based on static analysis of the structure subjected to lateral forces distributed over the building height according to *n*th modal shape. Chopra and Goel [51] showed that this procedure is accurate enough for practical application. The MPA procedure used in this work adopts the capacity spectrum method (CSM), a nonlinear static analysis procedure to assess the seismic vulnerability of buildings originally proposed by Freeman [52]. The procedure permits finding a correlation between earthquake ground motions and building performance [53] (ATC, 1982) comparing a response spectrum (representing structure demand) and a pushover curve (representing building capacity) by an iterative procedure. This latter aims to the definition of the performance point (PP), which represents the state of maximum inelastic displacement of a building for a given seismic event. To plot the two curves in the same chart, RS and pushover curves are transformed into an acceleration displacement response spectrum (ADRS). The whole procedure employed can be summarized in the following steps (see Figure 3):

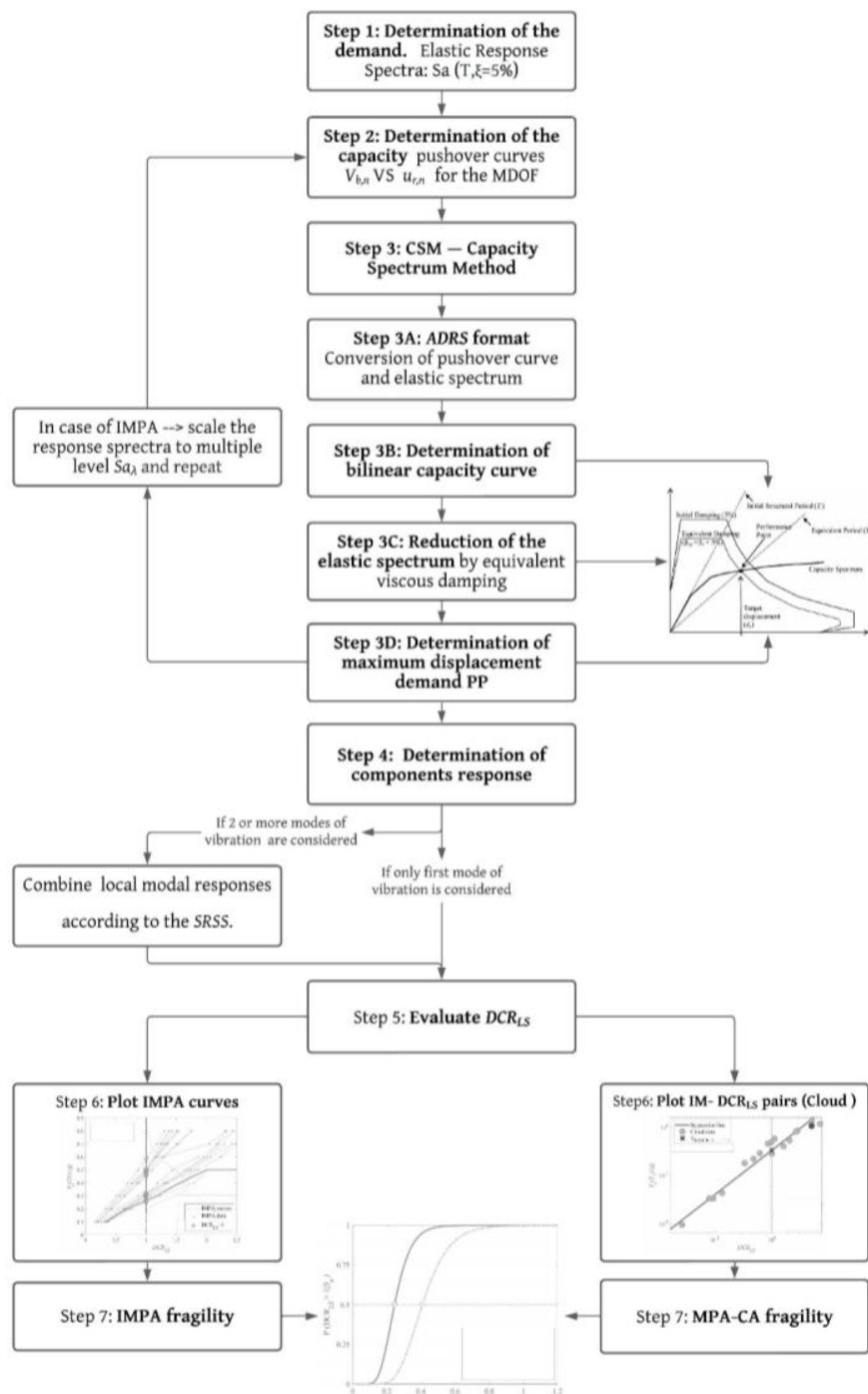

**Figure 3.** Flowchart of MPA-CA and IMPA procedures. $V_{b,n}$ and $u_{r,n}$ are the base shear and the top displacement respectively, PP is the performance point and $DCR_{LS}$ is the critical demand to the capacity ratio for the desired limit state (*LS*).

1. Determine demand: elastic response spectrum ($T$, $\xi$ = 5%);
2. Evaluate the capacity via pushover curves in terms of base shear $V_{b,n}$ versus top displacement $u_{r,n}$ for the MDOF structure subjected to lateral forces profile proportional to $n$th mode shape;
3. Determine maximum demand in terms of top displacement via the capacity spectrum method:

- Convert the pushover curve of the $n$th mode shape to a capacity curve in the ADRS format by (Equations (6)–(9)):

$$a_{C,n} = \frac{V_{b,n}}{M_{tot} \cdot \alpha_n} \qquad (6)$$

$$d_{C,n} = \frac{u_{r,n}}{\Gamma_n \cdot \phi_{n,r}} \qquad (7)$$

$$\Gamma_n = \frac{\phi_n^T M I}{\phi_n^T M \phi_n} \qquad (8)$$

$$\alpha_n = \Gamma_n \frac{\phi_n^T M I}{M_{tot}} \qquad (9)$$

where $M_{tot}$ is the total mass of the structure, $\phi_n$ is the $n$th natural vibration mode, $\phi_{n,r}$ is the amplitude of the $n$th natural vibration mode at the roof of the structure, and $\Gamma_n$ and $\alpha_n$ are the modal participation factor and modal mass of the nth mode, respectively;

- Convert 5% damped response spectrum from the standard pseudo-acceleration $S_a$ versus the period of vibration $T$ format to the ADRS format by (Equation (10)):

$$S_{De}(T) = S_a(T) \left( \frac{T}{2\pi} \right)^2 \qquad (10)$$

where $S_D(T)$ is the displacement spectrum;

- Plot demand and capacity diagrams together in the ADRS space. Determine the bilinear capacity curve. Iteratively determine the displacement demand for the $n$th mode shape. In this step, the dynamic analyses of a sequence of equivalent linear systems with successively updated values of equivalent viscous damping are involved;
- Reduce the elastic spectrum by the equivalent viscous damping (Equation (11));

$$\eta = \sqrt{\frac{10}{5 + v_{eq}}} \qquad (11)$$

- Determinate the performance point or the maximum expected demand in terms of top displacement;

4. Convert displacement demand found in step 3 to global top displacement and individual component of local deformation (i.e., interstory drift) for the $n$th mode shape;
5. Evaluate maximum demand to capacity ratio values according to Equation (1). If two or more modes of vibration are considered, combine the local modal responses according to the square-root-of-sum-of-squares (*SRSS*);
6. Estimate parameters of the linear regression model in the logarithm scale to fit the pairs of demand to capacity ratio ($DCR_{LS}$) and *IM*;
7. Draw structural fragility curve according to Equation (5).

### 2.3.2. Incremental Dynamic Analysis (IDA)

In IDA, a nonlinear structural model is subjected to a set of scaled ground motion records (accelerogram $a_\lambda$), each scaled to multiple levels of a monotonic scalable intensity measure such as $S_a$, PGA, PGV (herein $IM = S_a(T_1, \xi = 5\%)$). "As-recorded" unscaled time histories are scaled by using a non-negative scale factor ($\lambda$) to obtain a scaled accelerogram $a_\lambda$, in which amplitudes are scaled without changing the frequency content of signals. The output of the analysis is represented by a collection of IDA curves, which are a plot of the recorded $DCR_{LS}$ (DM) against $S_a(T_1, \xi = 5\%)$ (*IM*), all parameterized on the same *IM*s and DM [19].

Among all the analytical methods to develop the fragility based on IDA, the following is one of the simplest proposed [54]:

$$P(LS|IM = x) = P(DCR_{LS} \geq 1|IM = x) = P\left(IM^{DCR=1} \leq x\right) \tag{12}$$

In an EDP-based interpretation of the fragility (Equation (12)), the conditional probability of exceeding a limit state given an *IM*, herein the spectral acceleration, is equal to the probability of the demand to capacity ratio of exceeding 1 for a given Sa.

However, it is possible to express the fragility also as the complementary cumulative distribution function or "*IM*-based fragility" (Equation (13)). Incremental dynamic analysis is well suited to be represented by *IM*-based derivation of fragility. This interpretation expresses the seismic fragility as the probability of spectral acceleration values—denoted as $S_a{}^{DCR=1}$ and defined by intercepting all the IDA curves with the $DCR_{LS} = 1$—to be smaller than a given value. $DCR_{LS} = 1$ represents the threshold of a limit state *LS* and the intersection provides the empirical distribution of the random variable (*IM*), to which a model such as the lognormal appearing in Equation (12) can be fitted.

$$P\left(IM^{DCR=1} \leq x\right) = \phi\left(\frac{\ln x - \ln \eta_{S_a|DCR=1}}{\beta_{S_a|DCR=1}}\right) \tag{13}$$

In this Equation (13), $\phi$ denotes the standard normal (Gaussian) cumulative distribution (CDF) of two-parameters (median or log of mean $\eta$ and standard deviation $\beta$) estimated by the second-moment method or "METHOD A" (Equation (14)) proposed by Porter [54].

$$\ln \eta_{S_a|DRC=1} \cong \frac{\sum_{i=1}^{n} \ln S_a^{DCR=1}}{n} \qquad \beta_{S_a|DRC=1} \cong \sqrt{\frac{\sum_{i=1}^{n} \left(\ln S_{a,1}^{DCR=1} - \ln \eta_{S_a|DRC=1}\right)^2}{n-1}} \tag{14}$$

### 2.3.3. Incremental Modal Pushover Analysis (IMPA)

IMPA is a novel nonlinear static procedure proposed by Bergami and others first for buildings [40,55,56], and later adjusted also for bridges [57–59]. This procedure takes advantage of the simplicity of static analysis, but at the same time it grants the definition of the seismic demand for a certain range of intensity levels by scaling down response spectra. Conceptually, the procedure to find the maximum expected demand for the *j*th element of *l*th mechanism $D_{jl}$ is the same as that describe for MPA-CA in sub-Section 2.3.1, yet in IMPA procedure response spectra are scaled to multiple levels of a chosen monotonic scalable intensity measure as $S_a$, PGA, PGV (herein $IM = S_a(T_1, \xi = 5\%)$) (Figure 4). For each intensity level, the performance point (P.P.) can be determined and the demand measure $D_{jl}$ combined if two or more modes of vibration are considered, to define a multimodal $DCR_{LS}$. The output of the analysis can be represented as a collection of "IMPA" curves, which are a plot of the recorded $DCR_{LS}$ against *IM*, wholly similar to IDA curves. The intersection of IMPA curves and the chosen threshold of the limit state ($DCR_{LS} = 1$) generates the empirical distribution of the random variable (*IM*) for the probabilistic model of the fragility aforementioned in Section 2.3.1 (Equations (12) and (13)).

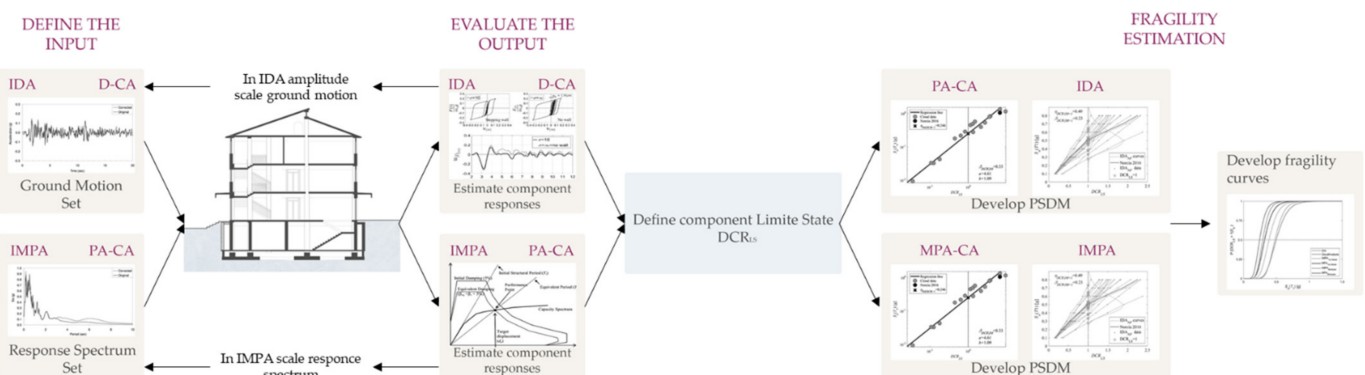

**Figure 4.** Flowchart schematically shows the main steps to develop seismic fragility curves for nonlinear static and dynamic procedures.

## 3. Numerical Application

### 3.1. Frame Description

The transverse frame modelled and analysed in this study comes from an actual school building in Norcia (Italy, 42.7941° North latitude, 13.0963° East longitude). The building, which originally consisted of three aligned blocks, was later joined into a single complex during the various refurbishment works. The building consists of a reinforced concrete (RC) frame structure with a footprint of 12.80 × 59.80 m and a maximum height (from the foundations), corresponding to the roof beams, of about 16 m. The building consists of a one-floor basement, a ground floor, three storeys and an attic above ground. The inter-story height is 3.50 m for the basement floor and ground floor, 3.30 m for the other three floors, and 2.5 m for the attic (Figure 5). Built in 1962, the school has survived a variety of seismic events before the 6.5 magnitude central Italy earthquake that caused severe damage to structural and especially non-structural parts in 2016. In accordance with the construction methods of the time of construction, the structure was designed using 2D models schematizing the reinforced concrete frames in the transverse direction of the building. Although the legislation of the time did not explicitly require it, the designer also took into account the seismic action by applying an acceleration of 0.07 g.

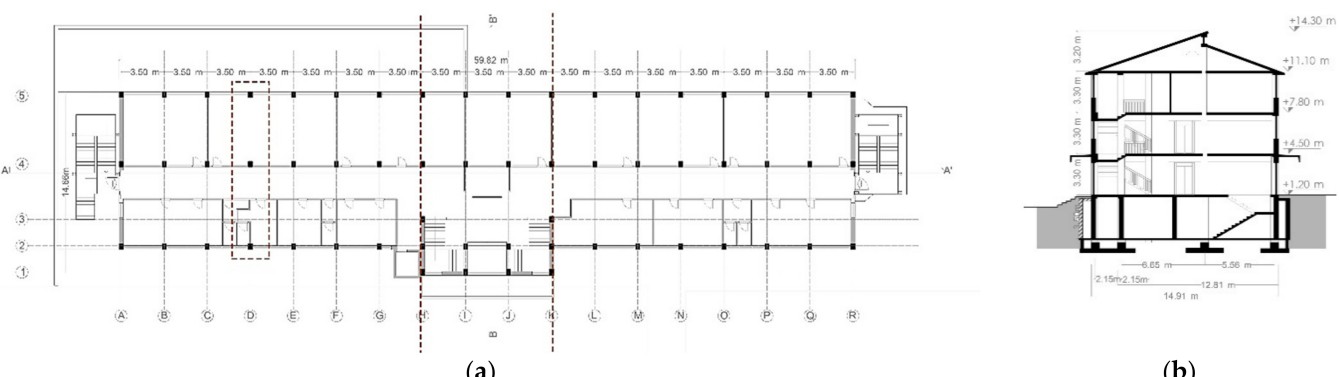

(a) (b)

**Figure 5.** *Cont.*

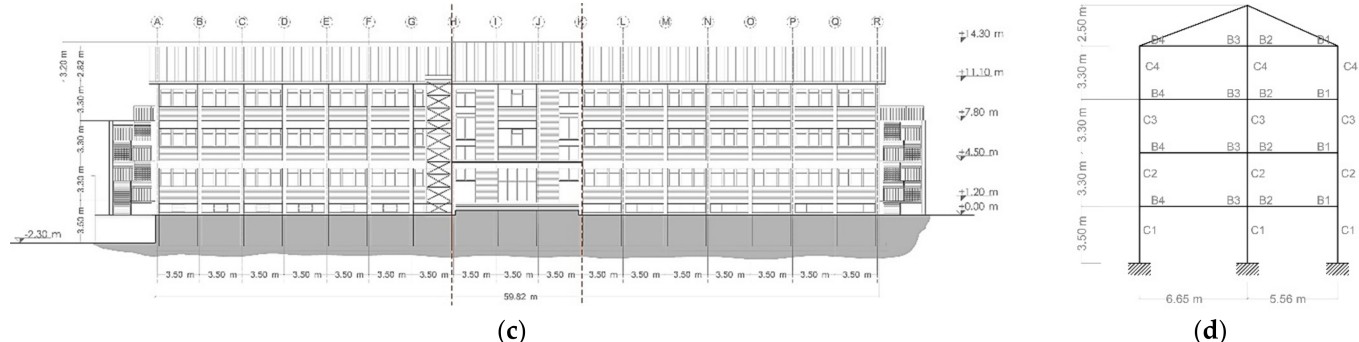

(c)                                                                    (d)

**Figure 5.** (**a**) Plan; (**b**) transverse section; (**c**) elevation; (**d**) modelled frame. The structural joint originally designed for thermal deformations are highlighted by orange hidden lines.

The frame under study belongs to one of the two lateral blocks (Figure 5) and is a two-bay (5.65 m and 5.56 m span) regular cross frame. It is considered to be fixed at the base, and basement and soil-structure interaction has not been considered.

The geometry of the frame, the column and beam geometry, and the reinforcement details are shown in Figures 5 and 6. The permanent structural load (G1) and permanent non-structural load (G2) are calculated as G1 + G2 = 5.1 kN/m$^2$ (from ground floor to 2nd floor), G1 + G2 = 4.1 kN/m$^2$ (3rd floor), G1 + G2 = 4.22 kN/m$^2$ (roof beams), the live load is taken as Q1 = 3 kN/m$^2$ (from ground floor to 2nd floor), Q1 = 1 kN/m$^2$ (3rd floor), Q2 = 1.8 kN/m$^2$ (roof beams) and taken as concentrated gravity on the columns at the edge of each floor. Each floor was assigned a seismic mass equal to 1/6 of the total mass of one of the three original blocks of the building.

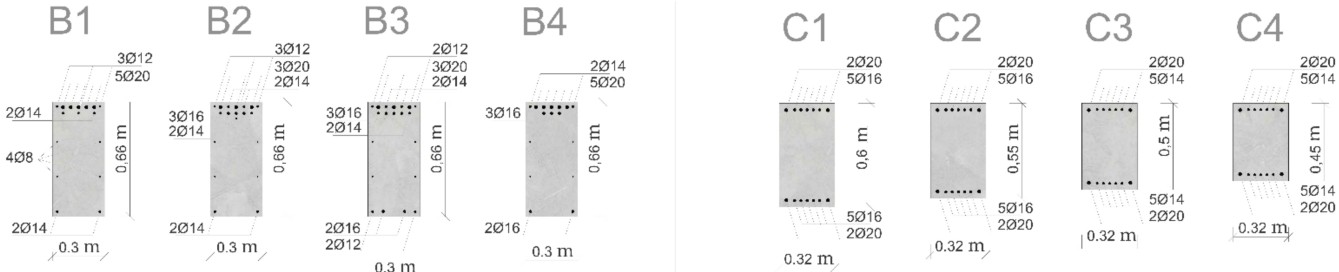

**Figure 6.** Columns (C) and beams (B) cross-sections and longitudinal reinforcement details.

Fundamental and second periods of the frame have been evaluated as $T_1$ = 0.62 s and $T_2$ = 0.21 s, respectively.

### 3.2. FE Model Description

The nonlinear FE model of the analysed frame was developed in the OpenSEES platform [60]. To account for the nonlinearity, "Beam With Hinges Element", already available in the OpenSEES library, was used to model columns and beams. This element adopts a lumped plasticity formulation with plastic hinges at the end of the element connected by an elastic link. This means that all nonlinearities are concentrated at the ends of the elements and can be only propagated along the length $L_p$ of the plastic hinge, unlike the distributed plasticity formulation where they may spread along the whole element. Therefore, the length of the plastic hinge $L_p$ plays an important role in avoiding the concentration of strain at the element ends. In this work, in good agreement with the $L_p$ evaluated by the equation proposed by Priestley and Park [61], it is assumed to be equal to the cross-section height. The two-point Gauss integration was used on the element interior, while two-point Gauss-Radau integration was applied over lengths of $4L_p$ at the element ends. A total of six integration points were used [62,63]. To account for non-linearity, a four-point moment-curvature relationship was assigned to the element

ends. The four-point law accounting for crack, yield, ultimate failure and collapse state (80% of ultimate failure) was evaluated through the software Response2000 [64], which can simulate nonlinear sectional behaviour by assuming a suitable law for the material. Due to the dependence of the sectional response on the applied axial load, it is assumed to be zero for beams, while it is estimated for the column considering their area of influence. Shear failure was not considered in the model. The Newton line search method was considered as the solution algorithm for the time-history analyses, which increased the effectiveness of the Newton–Raphson algorithm by introducing line search to solve the nonlinear residual equation. The tolerance and maximum number of iterations used were OpenSees default values [62]. Newmark integrator has been used and the convergence test was the normal displacement increment. The tolerance of the test is equal to $10^{-7}$ and the number of maximum iterations is 50. Rayleigh damping is adopted to account for energy dissipation.

In IDA, the scale factor $\lambda$ was chosen to scale the spectral acceleration at the fundamental period $S_a(T_1, \xi = 5\%)$, which was scaled to $IM = a_\lambda \in [0.1 \text{ g}, 0.8 \text{ g}]$ with $\Delta_{a\lambda} = 0.1$ g. Similarly, in IMPA, the response spectra were scaled multiple times to obtain scaled spectral acceleration at the fundamental period $S_a(T_1, \xi = 5\%)$ equal to $IM = S_{a\lambda} \in [0.1 \text{ g}, 0.8 \text{ g}]$ with $\Delta S_{a\lambda} = 0.1$ g. The mass damping coefficient and the stiffness damping coefficient of the Rayleigh damping are evaluated by considering the first and the second natural frequency of the case study. The percentage of critical damping is equal to 5%.

## 4. Results

### 4.1. Nonlinear Static Analysis Results

The pushover analysis has been performed in displacement control to reach a target displacement of 350 mm. The number of steps to reach the target displacement is 350 calculation steps with an increment of 1 mm each step. Figure 7 shows the capacity curves obtained applying two load distributions proportional to the first and second modal shapes, respectively. The capacity curve for the first load distribution reaches a maximum base shear of 492 kN corresponding to a top displacement of about 220 mm. The sequence of the plastic hinges activation with the relative calculation step is shown in Figure 8. A plastic hinge is activated when the reinforcement of the section reaches and exceeds the yield point: the first plastic hinges activated in the columns rather than in the beams, particularly in the upper stories which, according to Eurocode 8, have weak column–strong beam connections.

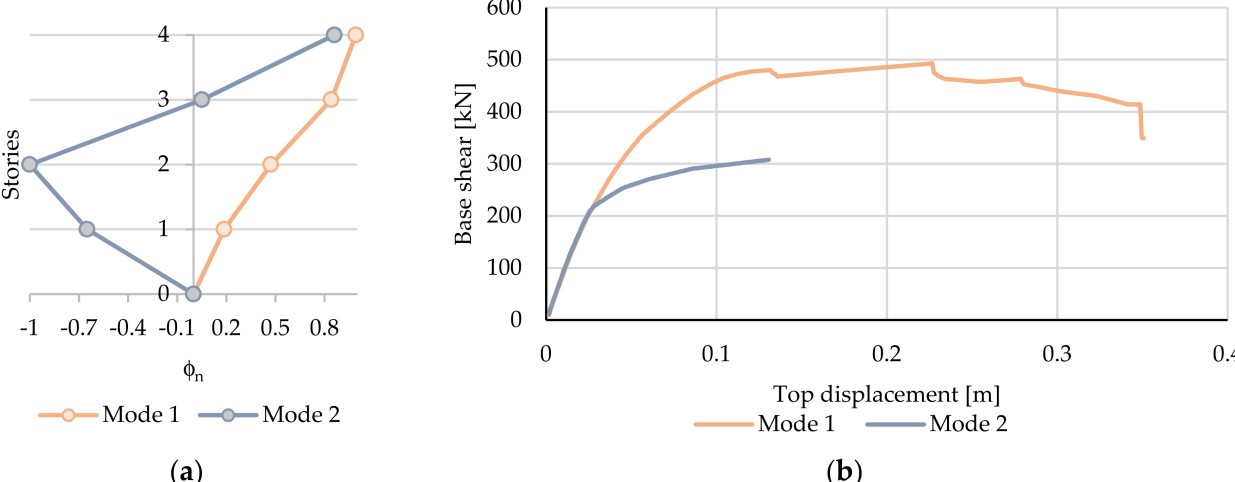

**Figure 7.** (**a**) First and second modal shapes and (**b**) capacity curves of the frame.

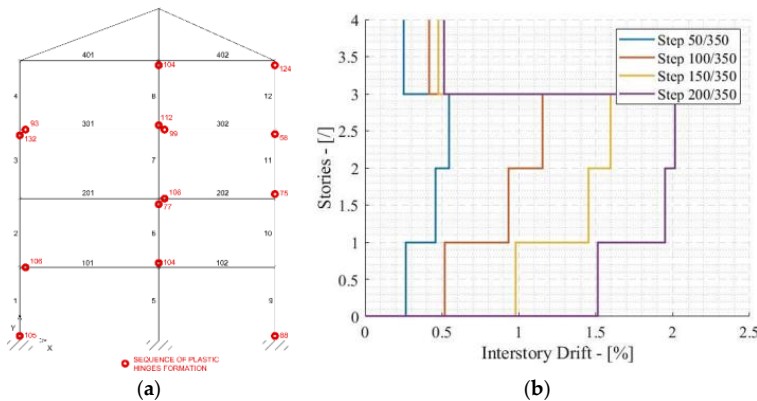

**Figure 8.** (**a**) Plastic hinges and (**b**) interstory drift at different steps of the pushover analysis.

Finally, Figure 8 shows the interstory drift at different steps of the analysis, with an interval of 50 steps. It shows a large concentration of interstory drift in the second and third stories, while the fourth one is moving almost rigidly.

### 4.2. Nonlinear Dynamic Analysis Results

A total number of six subsets of near-fault records and six subsets of far-field records were examined. The subsets were randomly generated from the main set of 210 records to comply with the general rules for record selection explained in Section 2.2. The mean and standard deviation of the normal distributions for magnitude, epicentral distance and spectral acceleration of each record selection were given in Table 2.

**Table 2.** Mean and standard deviation of each record selection magnitude, epicentral distance and spectral acceleration normal distributions.

|  | Sel. 1 | Sel. 2 | Sel. 3 | Sel. 4 | Sel. 5 | Sel. 6 | Sel. 7 | Sel. 8 | Sel. 9 | Sel. 10 | Sel. 11 | Sel. 12 |
|---|---|---|---|---|---|---|---|---|---|---|---|---|
|  | NF | NF | NF | NF | NF | NF | FF | FF | FF | FF | FF | FF |
| $\mu_{Rjb}$ | 4.84 | 4.31 | 4.73 | 4.63 | 4.56 | 4.28 | 20.80 | 19.55 | 19.32 | 20.89 | 19.66 | 18.84 |
| $\sigma_{Rjb}$ | 2.95 | 3.04 | 3.19 | 3.08 | 2.97 | 3.04 | 10.09 | 8.97 | 6.68 | 9.84 | 9.19 | 6.44 |
| $\mu_{Mw}$ | 5.83 | 5.88 | 6.03 | 5.86 | 5.92 | 5.85 | 5.73 | 5.80 | 5.64 | 5.72 | 5.54 | 5.72 |
| $\sigma_{Mw}$ | 0.72 | 0.77 | 0.71 | 0.82 | 0.82 | 0.82 | 0.75 | 0.73 | 0.59 | 0.60 | 0.56 | 0.60 |
| $\mu_{Sa}$ | 0.41 | 0.44 | 0.45 | 0.45 | 0.49 | 0.46 | 0.18 | 0.23 | 0.22 | 0.19 | 0.19 | 0.21 |
| $\sigma_{Sa}$ | 0.39 | 0.40 | 0.37 | 0.44 | 0.40 | 0.42 | 0.26 | 0.26 | 0.29 | 0.27 | 0.27 | 0.27 |

As expected, the results show that IDA and IMPA are less dependent on record selection, with the mean values of fragility curves ranging from 0.463 g to 0.525 g and from 0.387 to 0.432, respectively. In contrast, D-CA and MPA-CA show greater dependence on record selection, with mean values between 0.479 g and 0.724 g and between 0.321 g and 0.621 g, respectively (Figure 9). It can be pointed out that IMPA provides the most conservative results for all the datasets studied, as shown in Table 3. Moreover, this methodology seems to be the more accurate with respect to IDA in estimating vulnerability for the far-field record selections.

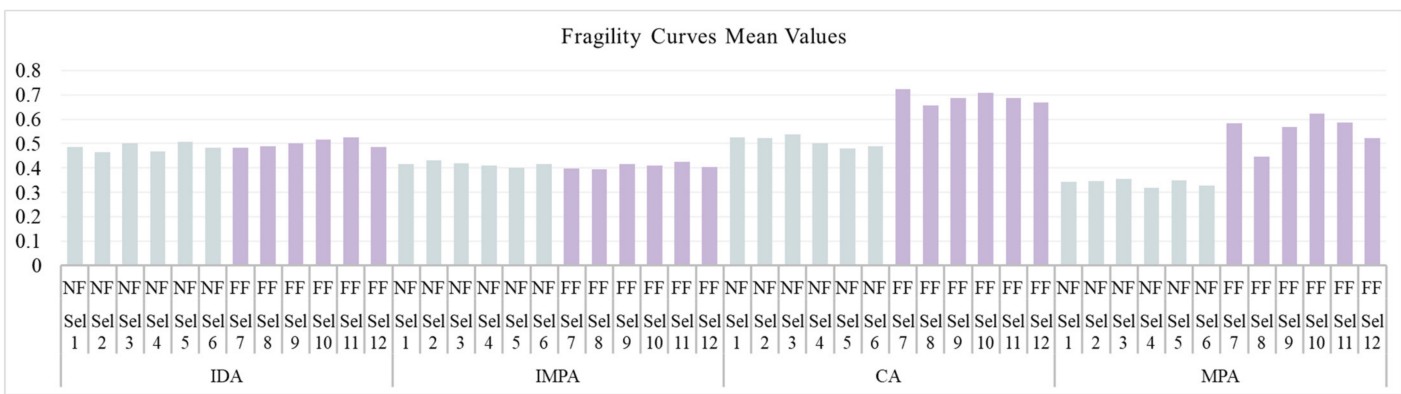

**Figure 9.** Histogram of mean values of fragility curves for different methodology and selection of records.

**Table 3.** Percentage variation of the 16th percentile, mean and 84th percentile of fragility curves for different methods with respect to IDA.

| Methodology | IDA | IMPA$_1$ | D-CA | MPA$_1$-CA | IDA | IMPA$_1$ | D-CA | MPA$_1$-CA | IDA | IMPA$_1$ | D-CA | MPA$_1$-CA | IDA | IMPA$_1$ |
|---|---|---|---|---|---|---|---|---|---|---|---|---|---|---|
| Fractile | 0.16 | 0.16 | 0.16 | 0.16 | 0.5 | 0.5 | 0.5 | 0.5 | 0.84 | 0.84 | 0.84 | 0.84 | $\beta$ | $\beta$ |
| | [g] | % | % | % | [g] | % | % | % | [g] | % | % | % | | |
| Sel. 1 $_{NF}$ | 0.388 | −20% | 1% | −34% | 0.487 | −15% | 8% | −29% | 0.602 | 15% | 44% | −5% | 0.23 | 0.29 |
| Sel. 2 $_{NF}$ | 0.353 | −13% | 3% | −33% | 0.464 | −7% | 12% | −25% | 0.609 | 32% | 62% | 8% | 0.27 | 0.35 |
| Sel. 3 $_{NF}$ | 0.396 | −21% | 7% | −38% | 0.500 | −16% | 8% | −29% | 0.632 | 12% | 38% | 1% | 0.24 | 0.29 |
| Sel. 4 $_{NF}$ | 0.354 | −15% | 5% | −46% | 0.466 | −12% | 8% | −32% | 0.615 | 20% | 45% | 15% | 0.28 | 0.31 |
| Sel. 5 $_{NF}$ | 0.399 | −30% | −11% | −42% | 0.506 | −21% | −5% | −31% | 0.641 | 14% | 28% | 3% | 0.24 | 0.37 |
| Sel. 6 $_{NF}$ | 0.382 | −19% | −4% | −40% | 0.482 | −14% | 1% | −32% | 0.608 | 15% | 34% | −4% | 0.23 | 0.29 |
| Sel. 7 $_{FF}$ | 0.401 | −27% | 43% | 8% | 0.481 | −18% | 50% | 21% | 0.577 | 12% | 90% | 63% | 0.18 | 0.30 |
| Sel. 8 $_{FF}$ | 0.376 | −26% | 32% | −28% | 0.488 | −19% | 35% | −9% | 0.634 | 14% | 78% | 50% | 0.26 | 0.35 |
| Sel. 9 $_{FF}$ | 0.398 | −23% | 44% | 3% | 0.501 | −17% | 37% | 14% | 0.631 | 13% | 64% | 58% | 0.23 | 0.31 |
| Sel. 10 $_{FF}$ | 0.402 | −30% | 46% | −2% | 0.515 | −21% | 37% | 21% | 0.659 | 16% | 65% | 91% | 0.25 | 0.38 |
| Sel. 11 $_{FF}$ | 0.429 | −26% | 34% | −3% | 0.525 | −19% | 31% | 12% | 0.643 | 7% | 57% | 58% | 0.20 | 0.29 |
| Sel. 12 $_{FF}$ | 0.387 | −22% | 47% | −1% | 0.485 | −17% | 38% | 8% | 0.607 | 11% | 62% | 49% | 0.23 | 0.29 |
| $\mu$ [g] | 0.39 | 0.30 | 0.47 | 0.31 | 0.49 | 0.41 | 0.60 | 0.45 | 0.62 | 0.56 | 0.76 | 0.65 | | |
| $\sigma$ | 0.02 | 0.01 | 0.10 | 0.09 | 0.02 | 0.01 | 0.10 | 0.12 | 0.02 | 0.02 | 0.09 | 0.17 | | |
| CoV | 0.05 | 0.05 | 0.21 | 0.29 | 0.04 | 0.03 | 0.16 | 0.27 | 0.03 | 0.04 | 0.12 | 0.27 | | |

The two subsets of records studied in-depth (Sel. 1 $_{NF}$ and Sel. 7 $_{FF}$ (Table 1)) comprise two different records of the real seismic event to which the case study was exposed in 2016, the 6.5 magnitude Central Italy earthquake, which, as mentioned, caused severe damage to structural and especially non-structural component of the school. Figures 10 and 11 shows IDA curves and IMPA curves: the curves of IDA referring to the Norcia 2016 earthquake exceed the threshold of $DCR_{LS}$ for the intensities of $S_a$ equal to 0.5 g and 0.57 g for NF and FF selection, respectively. The results of D-CA show that, as expected, the $DCR_{LS}$ threshold for the SLV limit state is exceeded for both the near-fault record and far-field record, with $DCR_{LS}$ values of 2.57 and 1.01, respectively (Figure 12). Similarly, even higher $DCR_{LS}$ are obtained from the cloud analysis based on nonlinear static analysis (Figure 13).

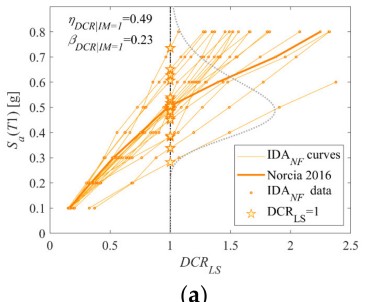
(a)

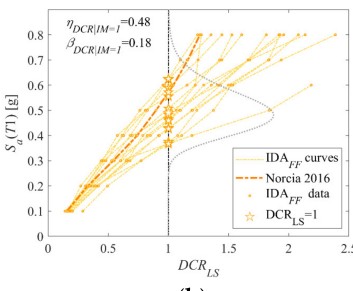
(b)

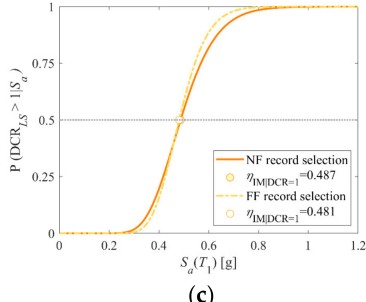
(c)

**Figure 10.** IDA: (**a**) comparison among IDA curves for near-fault record selection and (**b**) far-field record selection; (**c**) comparison between the two fragility curves obtained.

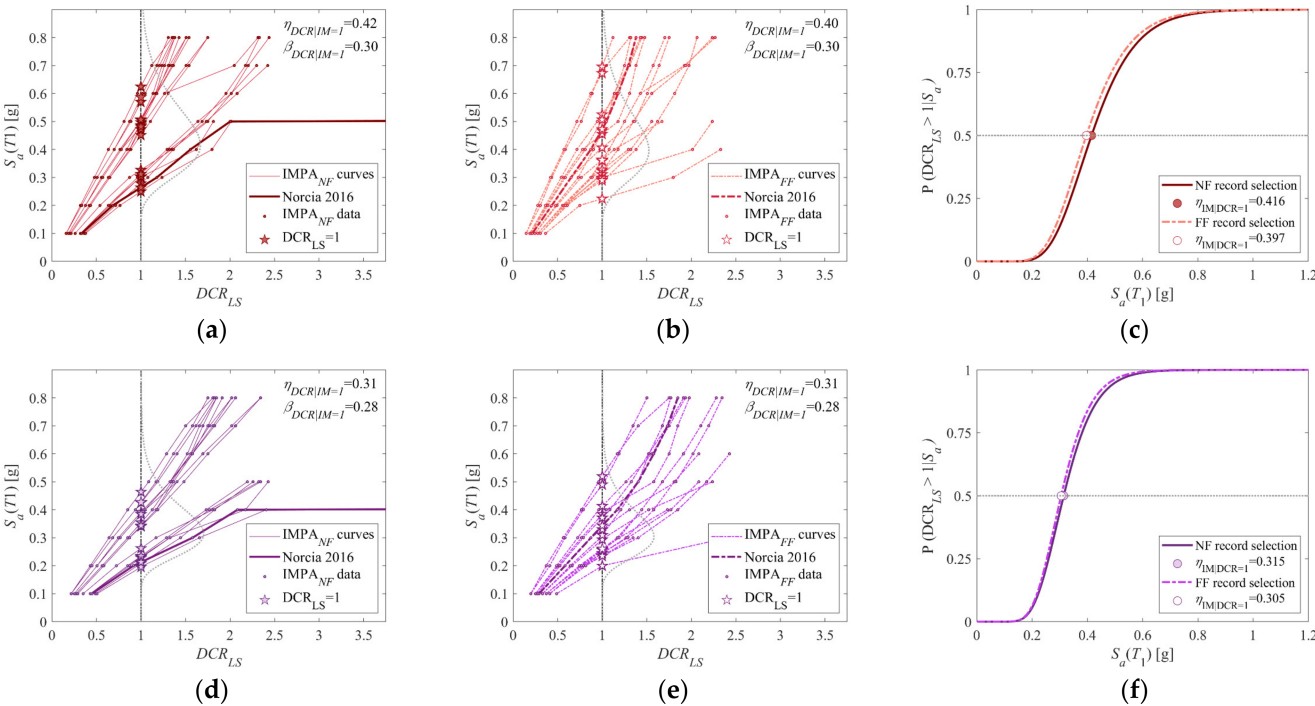

**Figure 11.** IMPA: (**a**) comparison among IMPA curves for near-fault record selection and (**b**) far-field record selection; (**c**) comparison between the two fragility curves obtained considering only the first mode; (**d**) comparison among IMPA curves for near-fault record selection and (**e**) far-field record selection; (**f**) comparison between the two fragility curves obtained considering the first mode and second mode.

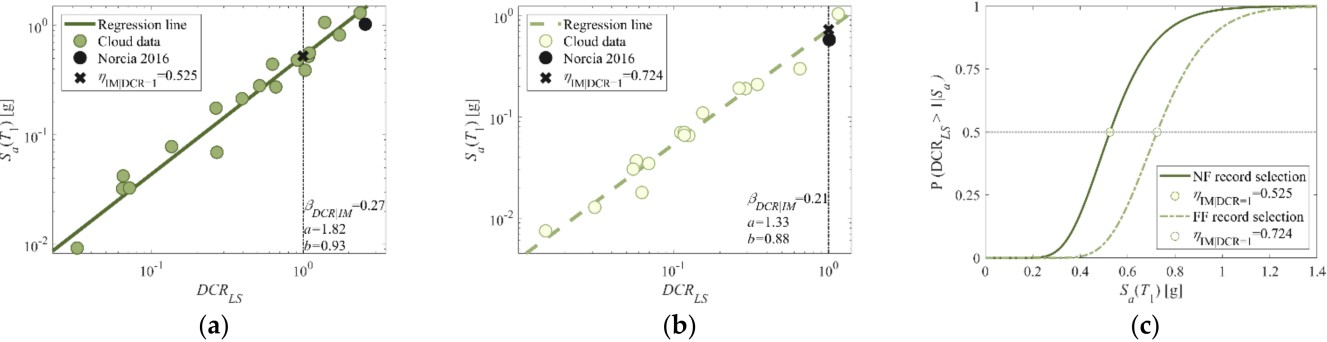

**Figure 12.** D-CA: (**a**) comparison among dynamic analysis cloud data regressions for near-fault record selection and (**b**) far-field record selection; (**c**) comparison between the two fragility curves obtained.

To check the consistency of modal and multimodal IMPA and pushover-based cloud, the developed fragility curves (Figure 14) are compared with those of IDA. The accuracy of the prediction of the different fragility models with respect to IDA is quantified by normalized root-mean-square deviation (*RMSD*). It is evaluated according to the following Equation (15), assuming that the values derived by Equation (12) (IDA) are the reference ones:

$$RMSD(\%) = \sum_{i=1}^{n} \sqrt{\frac{(\hat{y}_i - y_i)^2}{\hat{y}_i^2}} \tag{15}$$

where *n* is the number of points, and $y_i$. and $\hat{y}_i$ are the predicted and reference probability of exceeding the considered limit state (*LS*), respectively.

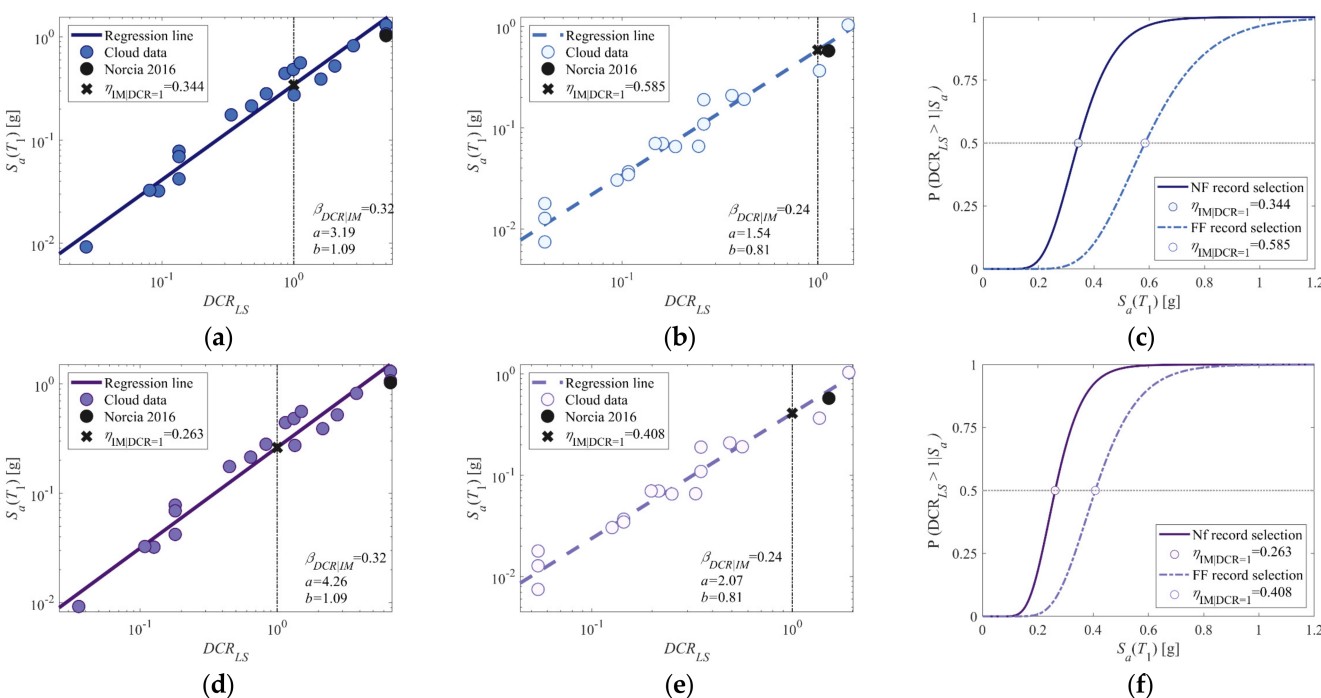

**Figure 13.** MPA-CA: (**a**) comparison among modal pushover analysis cloud data regressions for near-fault record selection and (**b**) far-field record selection; (**c**) comparison between the two fragility curves obtained considering only the first mode; (**d**) comparison among modal pushover analysis cloud data regressions for near-fault record selection and (**e**) far-field record selection, (**f**) comparison between the two fragility curves obtained considering the first and the second mode.

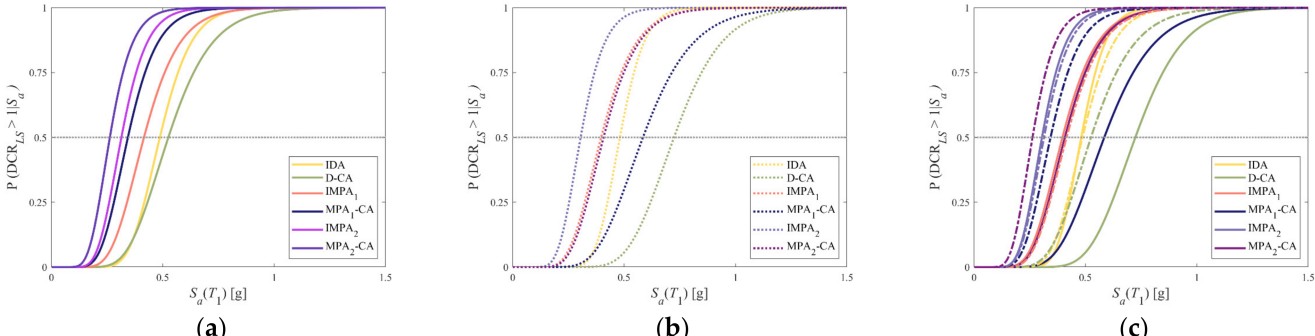

**Figure 14.** Comparison among fragility curves for all aforementioned methods: (**a**) comparison among fragility curves for near-fault (NF) selection (**b**) far-field (FF) selection, and (**c**) total.

The following Table 4 shows the comparison in terms of the percentage variation of the median, 16% and 84% fractiles of the fragility curves with respect to IDA and the absolute values of the standard deviation of each method. Regarding the selection of NF records, the cloud method appears solid in estimating the 50%, 16% and 84% fractiles compared to IDA with the smallest normalized root-mean-square deviation of 8%. However, when it comes to far-field records, IMPA appears to be the most accurate methodology for estimating fragility. The inclusion of two or more vibration modes in the assessment of the maximum multimodal $DCR_{LS}$ does not seem to be essential, as it leads to very conservative results.

**Table 4.** Percentage values represent the percentage change with respect to the values observed for IDA.

| Methodology | Near-Fault Record (Sel. 1) | | | | | Far-Field Record (Sel. 7) | | | | |
|---|---|---|---|---|---|---|---|---|---|---|
| | $\eta_{16\%}$ [g] | $\eta_{50\%}$ [g] | $\eta_{84\%}$ [g] | $\beta$ | *RMSD* | $\eta_{16\%}$ [g] | $\eta_{50\%}$ [g] | $\eta_{84\%}$ [g] | $\beta$ | *RMSD* |
| IDA | 0.388 | 0.487 | 0.612 | 0.22 | - | 0.481 | 0.401 | 0.577 | 0.18 | - |
| D-CA | 1% | 8% | 15% | 0.26 | 8% | 43% | 50% | 58% | 0.21 | 41% |
| MPA$_1$-CA | −34% | −29% | −24% | 0.33 | 11% | 8% | 21% | 36% | 0.24 | 20% |
| MPA$_2$-CA | −50% | −46% | −42% | 0.33 | 15% | −24% | −15% | −5% | 0.24 | 5% |
| IMPA$_1$ | −20% | −15% | −9% | 0.28 | 6% | −27% | −18% | −7% | 0.30 | 6% |
| IMPA$_2$ | −39% | −35% | −32% | 0.27 | 13% | −42% | −37% | −30% | 0.28 | 12% |

## 5. Conclusions

This paper compares fragility curves obtained by various known static and dynamic nonlinear procedures. Incremental modal pushover analysis (IMPA) is proposed as an alternative to IDA, which is currently considered the most reliable method, to determinate IMPA curves and thus seismic fragility. Similarly, both MPA and NL-THA are used to determine capacity in the well-known cloud method. For this study, it was necessary to perform a relatively small number of nonlinear time histories using two different data sets. These differ in the range of Joyner-Boore distance ($R_{jb}$) and are scattered in a range of magnitude.

The following conclusions can be drawn from this study that is limited to a simplified 2D frame model and a small set of records:

- Comparison of fragility curves shows that, in the case of methodologies distinguished by scaling (in terms of accelerograms (IDA) or response spectra (IMPA)), near-fault records and far-field record selections have led to nearly equivalent results. In contrast, the results in terms of fragility when using records without scaling, i.e., in D-CA and MPA-CA, show clear differences in the whole range of intensities;
- Fragility curves that only consider the contribution of the first mode in determining $DCR_{LS}$ have led to more accurate results in relation to IDA, so the inclusion of higher mode contributions does not seem to be essential for low to medium buildings (up to nine stories [20]);
- A total number of 12 subsets have been extracted from the main 210 set of records and exanimated, but results are not fully reported in this paper. The results have shown that D-CA leads to a smaller vulnerability than MPA-CA and IMPA in all selection and for the whole range of intensities. Methodologies based on the pushover analysis, on the contrary, have led to more conservative results, especially for 16% and 50% fractiles;
- IDA shows less sensitivity to record-to-record variability. It should be noted, however, that IMPA, despite its slightly greater sensitivity, has the advantage of a large reduction in the computational effort required to perform the structural analysis. In IMPA, the total time required relates mostly to the post-processing phase, which is no different for small 2D frames or more complex 3D buildings.

A more comprehensive validation is needed to confirm the obtained results and draw more general conclusions.

**Author Contributions:** Conceptualization, C.P.C., A.P., C.N. and B.B.; methodology, C.P.C., A.P. C.N. and B.B.; software, C.P.C. and A.P.; formal analysis, C.P.C. and A.P.; data curation, C.P.C. and A.P.; writing—original draft preparation, C.P.C. and A.P.; writing—review and editing, C.P.C., B.B., A.P. and C.N.; visualization, C.P.C. and A.P.; supervision, C.N. and B.B. All authors have read and agreed to the published version of the manuscript.

**Funding:** The authors gratefully acknowledge the funding received by The Laboratories University Network of Seismic Engineering (ReLUIS): research project ReLUIS/DPC 2019–2021 Reinforced Concrete Existing Structures.

**Institutional Review Board Statement:** Not applicable.

**Informed Consent Statement:** Not applicable.

**Data Availability Statement:** Publicly available datasets were analyzed in this study. This data can be found here: https://ngawest2.berkeley.edu/spectras/new?sourceDb_flag=1; https://www.seismosoc.org/Publications/BSSA_html/bssa_104-5/2013191-esupp/index.html (all accessed on 15 February 2022).

**Conflicts of Interest:** The authors declare no conflict of interest.

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
