# Peer review of "IMPA versus Cloud Analysis and IDA: Different Methods to Evaluate Structural Seismic Fragility"

_applsci, doi:10.3390/app12073687_

Round 1

Reviewer 1 Report

Please see comments in pdf file.

Author Response

Thank you for taking the time to review and comment upon our manuscript “ IMPA versus Cloud Analysis and IDA, different methods to evaluate structural seismic fragility”. The authors found the advice constructive and have incorporated the suggestions into the revised paper, as reported in the following point by point reply to comments:

General concept comments

Lack of accuracy about the objective of the paper must be avoided. It is strongly recommended that authors decide which is the novel aspect of their research and stress it in the title, abstract, main text, and conclusions; other findings must be presented as secondary. There are two main concepts in the text: one is the novel application of IMPA to derive fragility curves; the other is the effect of record scaling in the fragility curves and the comparison between different methods. The fact that the current study is the first to propose IMPA to derive fragility curves [lines 73-74] is suggested but not clearly stated. This ambiguity must be removed. If this study is the first, this should be emphasized as a significant contribution. If it is not the first, it should be clearly stated, and previous studies on the subject must be cited. Indeed, if the target of the paper is to introduce IMPA to obtain fragility curves for the first time, the main title of the article and the abstract should be changed to stress this fact. In this context, please consider Lines 422-423, where it is stated that ‘the primary objective of the paper is to verify the effect of scaling on different bin of records’; the study is not exactly focused on this objective. Is this the main finding of the paper? Is it a new finding or a known fact? Please indicate if previous studies treated the effect of scaling, and if the results are coincident.

The primary purpose of this study is to understand whether IMPA is an appropriate procedure for constructing fragility curves. Considering that the procedure has proved to be fast and does not have a large computational demand (see Bergami et al.), the authors think this may be suggested as a simpler alternative to known methodologies like IDA. “This paper aims to evaluate the reliability of structural fragility derived by the methods mentioned above, advancing an IM-based derivation of structural fragility, strikingly similar to IDA, based on IMPA. It’s known that IDA has a small sensitivity to record-to-record variability compared to other methodologies. Nevertheless, the results indicate that, amidst its slightly higher sensitivity, IMPA has the advantage of requiring considerably smaller computational effort to perform the structural analysis.  The authors argue that scaling response spectra at a higher range of intensities might introduce less uncertainties than a simple amplitude scaling of ground motions. Further steps of this research will address how the uncertainties in the seismic input affect the reliability of IMPA vs IDA seismic fragility for strong ground motions.” Lines 95-104

  • If possible, please include values of significant duration (time span in which the Arias Intensity for the motion changes from 5% to 95%) in the ground motion description (Table 1 and Table 2). Significant duration presents a strong correlation with damage. Other measures of duration would be acceptable.

Thank you for this helpful comment. We have only added the duration values 5-75% and duration 5-95% to Table 1. Unfortunately, since not all the data were available, these authors could not include them in Table 2.

  • The type of failure mechanism considered [lines 106-107] is not clear. There are many ductile mechanisms in a frame. Do the authors mean ‘complete mechanism’? Are ‘partial 3 mechanisms’ (which might produce collapse) not included? This important paragraph is obscure and needs further clarification.

In this paper, the only potential failure mechanism considered is a ductile failure mechanism in the columns and beams. In the deformation-based critical DCRLS, the demand D is expressed in terms of maximum chord rotation for the component. The capacity C is the component chord rotation capacity is evaluated according to the equation. (Lines 110-116).However, the weakest link formulation of the critical demand-to-capacity ratio used is general with respect to the failure mechanisms considered. The authors acknowledge that this is an important simplification so that future studies will definitely be considered a greater number of mechanisms, including global mechanisms, soft-story, etc.

  • Some additional information on the frame is necessary: i) is the frame weak columns strong beam? Please discuss ratios of column-to-beam bending strength and include moment-rotation diagrams of the cross sections presented; ii) first and second modal pushover curves of the main frame in the study must be included, with and without normalization to mass and yield displacement, and showing interstorey drift to storey shear curves for all stories, which is important to verify partial collapse mechanisms that may invalidate study results; iii) a schematic plot showing plastic hinges for the collapse mechanism would be valuable.

Thank you for this helpful comment. According to Eurocode 8 - Chapter 5 the frame has a strong column-weak beam for the lower stories (1st and 2nd), conversely, the upper stories have a weak column-strong beam. In Figure XX is shown the formation of the plastic hinges, it is clear that the first five plastic hinges are located on the upper column. This makes it suitable to consider just the column in the definition of the frame fragility.

As suggested, we included this information in the frame description, see Lines xxxx.

The authors also add the manuscript some figures:

Figure 7: shows the first and second modal push-over curves;

Figure 8a: shows the activation of the plastic hinges

Figure  8b: shows  the interstorey drift for all stories.

  • The four methods are introduced using acronyms (IDA, IMPA, D-CA, MPA-CA). However, these acronyms are not consistently used in the text. For instance, Table 3 uses ‘CA’ and ‘MPA’, instead of ‘D-CA’ and ‘MPA-CA’. This inconsistency should be removed. Please consider including a consistent description of the type ‘MPA-CA-1’ or ‘MPA-CA-2’ to specify if one or two modes are considered (also for IMPA).

We appreciate your noting and reporting these inconsistencies in this document. We have modified our text and images to ensure acronyms are appropriate.

  • Please clarify the inherent damping ratio adopted for the analysis (5%?) and the periods for which adjustment of Rayleigh damping was performed.

Thank you for this comment. In our model, mass damping coefficient and stiffness damping coefficient in the equation of the Rayleigh damping are evaluated by considering the first and the second natural frequency of the case study. The percentage of critical damping is equal to 5%.  This is been now clarified in Line 343-345.

  • Figure 7 and Table 3 give an excellent overview of mean results; it would be desirable in both Table and Figure to include dispersion results for DCRLS = 1 where possible (IDA and IMPA).

The authors appreciate this suggestion. Now Table 3 includes the values of the dispersion of the DCR for both methodologies. The authors believe that adding other information to Figure 7 would make this latter hard to read.

Finally, the Authors would like to thank again all the reviewers for their important comments, including all the specific comments, hoping that all the comments have been addressed in the best way.

Reviewer 2 Report

This is a study on the Fragility Estimation of structures using some available and new approaches. The manuscript investigates a very valuable and trending topic, and it has the potential to be published in Applied Sciences. However, there are some comments to be responded to before a decision can be made:

1. Literature review needs to be extended with more relevant studies. Some are listed below:

-Mohamed Nazri, F., Miari, M. A., Kassem, M. M., Tan, C. G., & Farsangi, E. N. (2019). Probabilistic evaluation of structural pounding between adjacent buildings subjected to repeated seismic excitations. Arabian Journal for Science and Engineering44(5), 4931-4945.

-Yang, T. Y., Farsangi, E. N., & Tasnimi, A. A. (2016). Influence of concurrent horizontal and vertical ground excitations on the collapse margins of non-ductile RC frame buildings. Structural Engineering and Mechanics, An Int'l Journal59(4), 653-669.

-Kassem, M. M., Nazri, F. M., & Farsangi, E. N. (2020). The seismic vulnerability assessment methodologies: A state-of-the-art review. Ain Shams Engineering Journal11(4), 849-864.

-Farsangi, E. N., Tasnimi, A. A., & Mansouri, B. (2015). Fragility assessment of RC-MRFs under concurrent vertical-horizontal seismic action effects. Computers and Concrete16(1), 99-123.

-Kassem, M. M., Nazri, F. M., & Farsangi, E. N. (2020, February). The efficiency of an improved seismic vulnerability index under strong ground motions. In Structures(Vol. 23, pp. 366-382). Elsevier.

-Farsangi, E. N., Takewaki, I., Yang, T. Y., Astaneh-Asl, A., & Gardoni, P. (Eds.). (2019). Resilient structures and infrastructure. Berlin, Germany: Springer.

-Dehghani, S., Fathizadeh, S. F., Yang, T. Y., Farsangi, E. N., Vosoughi, A. R., Hajirasouliha, I., ... & Takewaki, I. (2021). Performance evaluation of curved damper truss moment frames designed using equivalent energy design procedure. Engineering Structures226, 111363.

-Rakicevic, Z., Bogdanovic, A., Farsangi, E. N., & Sivandi-Pour, A. (2021). A hybrid seismic isolation system toward more resilient structures: shaking table experiment and fragility analysis. Journal of Building Engineering38, 102194.

2. The authors have chosen the first-mode spectral acceleration Sa(T1,ξ=5%) as the IM in their study. A clarification is needed; why is a more reliable IM (e.g., Vector IM) not used?

3. The authors should clarify if the selected Near-fault records include Pulse-Type records as well? If yes, they should be separated from the non-pulse ground motions.

4. Why didn’t the authors implement the Conditional Mean Spectrum to match and scale the ground motions?

5. Figs. 3 and 4, if taken from an external source, should be referenced.

6. Has there been any model verification? If yes, the verification results should be presented in the revised version.

7. The results in Figs. 8 and 9 are not very clear and readable. They should be improved in the revised version.

8. It is suggested that the authors add a new section and calculate the Collapse Margin Ratios (CMRs) based on FEMA P695 methodology.

Author Response

Thank you for taking the time to review and comment upon our manuscript “ IMPA versus Cloud Analysis and IDA, different methods to evaluate structural seismic fragility”. The authors found the advice constructive and have incorporated the suggestions into the revised paper, as reported in the following point by point reply to comments:

  1. Literature review needs to be extended with more relevant studies.
  2. The authors thank the reviewer for pointing this out. The authors agree with this comment and have reviewed and extended the literature, including the suggested ones.
  3. The authors have chosen the first-mode spectral acceleration Sa(T1,ξ=5%) as the IM in their study. A clarification is needed; why is a more reliable IM (e.g., Vector IM) not used?
  4. The reviewer has raised an important point. The authors are aware that several IM has been proposed and investigated in the literature. For example, the use of 65 different IMs is discussed in “Mackie, K. R., & Stojadinović, B. (2005). Fragility basis for California highway overpass bridge seismic decision making. Pacific Earthquake Engineering Research Center, College of Engineering, University of California, Berkeley”. This study suggested that spectral-acceleration at the first-mode period (and spectral displacement (Sd) too) is an ideal IMs as they found its use to reduce uncertainty in the demand models. Other IM, e.g., the Vector IM (add reference), are very useful and interesting, but for simplicity and according to the scope of this research the first-mode spectral acceleration has been proposed. Other IM will be investigated in future research steps
  5. The authors should clarify if the selected Near-fault records include Pulse-Type records as well? If yes, they should be separated from the non-pulse ground motions.
  6. Thank you for this useful comment. Our previous selection of Near Fault Record included two Impulsive signals. The latter have been identified using the open-source algorithm proposed by Shahi and Baker. The authors excluded these records from the selections and replaced them with non-pulse ground motions.
  7. Why didn’t the authors implement the Conditional Mean Spectrum to match and scale the ground motions?
  8. We acknowledge that record selection might have been done to match the CMS, but we choose to select the records in accord to what Jalayer and her coauthors have proposed in “Bayesian Cloud Analysis: efficient structural fragility assessment using linear regression” and other works by same authors referenced if our manuscript. However, we thank the reviewers for the suggestion because we recognize that this paper represents only a first step in this research. Many other aspects should be considered and investigated.
  9. Figs. 3 and 4 should be referenced if taken from an external source.
  10. The Authors thank the reviewer for the useful comment. Figs. 3 and 4. have been prepared by the Authors and not taken from external sources.
  11. The results in Figs. 8 and 9 are not very clear and readable. They should be improved in the revised version.
  12. As suggested by the reviewer, Figs. 8 and 9 sizes have been increased.
  13. It is suggested that the authors add a new section and calculate the Collapse Margin Ratios (CMRs) based on the FEMA P695 methodology.
  14. The Authors thank the reviewer for the important comment. The authors agree that the addition of a new section to include the calculation of CMRs would be an improvement to the manuscript, however, it is out of the scope of this paper and will be addressed in future research steps.

Thank you again for your thoughtful comments.

Reviewer 3 Report

  1. Introduction need to rewrite; some sentences are too long which is conveying the meaning what authors want to say.
  2. In line no. 30; authors’ have written …… among these, PEER (Pacific Earthquake Engineering Research Center) analysis methodology consists mainly in four stages: hazard analysis, structural analysis, damage analysis, and loss analysis [1]……. authors’ should discuss other methods before PEER methodology…………. Authors’ have mentioned PEER is as Centre “as abbreviated Pacific Earthquake Engineering Research Center” …….. then explain what is Pacific Earthquake Engineering Research Center analysis methodology………….… or it should be rewritten as: among these methodology, the methodology proposed by Pacific Earthquake Engineering Research Center (PEER) consists mainly four stages: hazard analysis, structural analysis, damage analysis, and loss analysis [1]……… Grammar should be checked thoroughly.
  3. In line no. 35; authors’ have written ……. First attempt to determinate fragility curves can be dates back to 1975 when the Seismic Design Decision analysis (SDDA) procedure was proposed in US [2]. ………….. “it should be written in a simple sentence like …………. The determination of fragility curve was firstly proposed by Whitman et al. [2] ……………. Check the sentences grammatically.
  4. In line no. 42; write limitations in place of restrictions and see the sentence.
  5. Line 46-50; it’s a very long sentence………. Make it simpler
  6. Many abbreviations are not defined, MSA, MPA, IMPA ……………… check it
  7. In Line 93; how come chapter 3?
  8. In Line 110; how come chapter 8?
  9. In line 112, what is this equation 7.2.5 of the Commentary?
  10. Remove the word, below, here, herein, left, right words and, Check the sentences grammatically.
  11. Section 2.2 should be rewritten and check the sentences grammatically. What magnitude (i.e., Mw, Mb, Ms) is reported in the table.
  12. Section 2.3 should be rewritten and check the sentences grammatically.
  13. Section 3.1, figure 1 does not show attic. What is 1960’s (1962) ……… Check the decimal points ………….. statement nowhere matching with fig. 2.
  14. Authors’ might have given a very good effort to find the problems, to analyze the results and to write this article………….. but it is difficult to understand the message which authors’ want to convey. further it disconnects the interest to read.  Kindly check the article thoroughly before submitting any journal.
  15.  

Author Response

Thank you for taking the time to review and comment upon our manuscript “ IMPA versus Cloud Analysis and IDA, different methods to evaluate structural seismic fragility”. The authors found the advice constructive and have incorporated the suggestions into the revised paper, as reported in the following point by point reply to comments:

  1. Introduction need to rewrite; some sentences are too long which is conveying the meaning what authors want to say.
  2. The authors thank the reviewer for pointing this out. Very long sentences have been shortened and/or rewritten to make the introduction more readable and clearer.
  3. ……..authors’ should discuss other methods before PEER methodology……
  4. The authors thank the reviewer for his helpful comment. In the revised paper, others PBEE methodologies (e.g., FEMA 273 (1997)) have been introduced and referenced. (see Lines 31-36).
  5. Line 46-50; it’s a very long sentence………. Make it simpler
  6. The authors thank the reviewer for his helpful comment. The sentence in Lines 46-50 has been modified and made simpler. V

“From then on, several methods to estimate fragility (expert-based, experimental, analytical, hybrid, empirical) have been developed by researchers worldwide, relying on different assumptions and restrictions to overcome prevalent intrinsic uncertainties.”

  1. Many abbreviations are not defined, MSA, MPA, IMPA ……………… check it
  2. The Authors thank the Reviewer for the useful comment. In the revised paper, all the abbreviations have been defined before using them.

3,5,7-10 Comments 3, 5 and from 7 to 10.

  1. The Authors thank the Reviewer for the useful comment. Lines 30, 35, 93, 110, 112 have been corrected. The following words have been removed were possible: below, here, herein, left, right.
  2. Section 2.2 should be rewritten and check the sentences grammatically.
  3. Section 2.3 should be rewritten and check the sentences grammatically.
  4. The Authors thank the Reviewer for the useful comment. The authors agree that the sentences should be improved. Grammatical errors have been corrected and many sentences are partly rewritten.
  5. … is difficult to understand the message which authors’ want to convey… kindly check the article thoroughly before submitting any journal.
  6. The Authors thank the Reviewer for the useful comment. The authors have carefully revised the paper to improve readability and to be clearer. Moreover, the authors hope that the main contribution of this manuscript to the research in the field is now clearly stated.

Finally, the Authors would like to thank again all the reviewers for their important comments and hope that all the comments have been addressed in the best way.

Reviewer 4 Report

Title: Experimental investigation on seismic performance of RC hill-side stilted buildings affected by vertical stiffness irregularity

Authors: Ruifeng Li; Liping Liu; Yingmin Li

General comment

In this study, the three nonlinear procedures (i.e., IDA, IMPA, and Cloud Method) were used to derive the drift demand of a school building, and the accuracy of three different procedures was investigated in a probabilistic domain (i.e., fragility curve). The accuracy and efficiency of IMPA compared to IDA were highlighted. The study well reports the simulation results related to the derivation of seismic fragility curves and the manuscript is well written. I would encourage the authors to resubmit the paper addressing the following minor comments.

Technical comment

  1. First of all, the resolution of figures, especially figures 3, 4, 7, 8, 9, 11, 12, should be increased to clearly visible of the text or the graphs in those figures.
  2. Introduction. The background of the derivation of the analytical fragility curve is well described referring to comprehensive literature. Additionally, please clearly state what a new contribution of this manuscript is.
  3. Line 153 of Page 6. What does the subscript LS in DCRLS mean? Is LS a life safety? If so, it is necessary to explain why life safety was decided as the damage measure. Also, please modify the DCRLS in Line 214 of Page 7.
  4. Line 288 of Page 10. Why the model was simulated in a transverse direction (i.e., shorter direction) rather than a longitudinal direction (i.e., longer direction)? Please explain the rationale for deciding the simulated direction.
  5. Conclusions. The meaning of Third bullet in the conclusion is ambiguous. Please clearly state with evidence for the basis of “D-CA lead to a greater vulnerability with respect to MPA-CA and IMPA”.

Author Response

Thank you for taking the time to review and comment upon our manuscript “ IMPA versus Cloud Analysis and IDA, different methods to evaluate structural seismic fragility”. We found the advice constructive and all spelling and grammatical errors pointed out by the reviewers have been corrected. We have responded to some of your comments individually below:

  1. First of all, the resolution of figures, especially figures 3, 4, 7, 8, 9, 11, 12, should be increased to clearly visible of the text or the graphs in those figures.
  2. Thank you for pointing this out. As suggested, we increased the quality of the pictures and also we have tried to increase the font size in all pictures as much as possible.
  3. Introduction. The background of the derivation of the analytical fragility curve is well described referring to comprehensive literature. Additionally, please clearly state what a new contribution of this manuscript is.
  4. We agree with the reviewer that the motivations behind this work, the main objectives and the limitations of the research are not fully described in the introduction. As suggested also by other reviewers, we have changed the introduction others PBEE methodologies, clarifying the objective and rewriting some sentences.
  5. Line 153 of Page 6. What does the subscript LS in DCRLS mean? Is LS a life safety? If so, it is necessary to explain why life safety was decided as the damage measure. Also, please modify the DCRLS in Line 214 of Page 7.
  6. The subscript LS in DCRLS means Limit State as indicated in Line 97 of the manuscript. This naming “LS” ( as Damage State DS) meant to be somehow general. However, in this paper the capacity of the structural components is evaluated in accordance with the equations provided by the Italian Code for a particular performance level, the life-safety limit state. As stated in “Record‑to‑record variability and code‑compatible seismic safety‑checking with limited number of records” [Jalayer et al, 2021], the weakest link formulation used herein for finding the critical demand to capacity ratio is suitable for safety checking for other limit states, although with the necessary precautions.
  7. Line 288 of Page 10. Why the model was simulated in a transverse direction (i.e., shorter direction) rather than a longitudinal direction (i.e., longer direction)? Please explain the rationale for deciding the simulated direction.
  8. It was considered a frame in the transverse direction because, at the time the building was built, the methodology adopted to design an RC structure consisted in schematizing it in a series of 2D RC frames arranged along a “principal” direction. This means that frames arranged in the longitudinal direction were not designed to give a proper contribution to the horizontal resistance of the building. The rationale for deciding the simulated direction is now explained in the manuscript Lines 316-320.
  9. Conclusions. The meaning of Third bullet in the conclusion is ambiguous. Please clearly state with evidence for the basis of “D-CA lead to a greater vulnerability with respect to MPA-CA and IMPA”.
  10. Thank you for noticing this mistake. The authors modified if the original statement of the third bullet in the conclusions was not clear. In fact, “the results have shown that D-CA leads to a smaller vulnerability than MPA-CA and IMPA in all selection and for the whole range of intensities.”

We appreciate all the suggestions and want to thank the reviewers for these insightful and useful comments.